# Identification of the centrosomal maturation factor SSX2IP as a Wtip-binding partner by targeted proximity biotinylation

**Alice H. Reis**[ID]◉, **Bo Xiang**¤◉, **Olga Ossipova, Keiji Itoh, Sergei Y. Sokol**[ID]*

Department of Cell, Developmental and Regenerative Biology, Icahn School of Medicine at Mount Sinai, New York, NY, United States of America

◉ These authors contributed equally to this work.
¤ Current address: Cancer Research Institute, Central South University, Changsha, Hunan, China
* sergei.sokol@mssm.edu

**Data Availability Statement:** All relevant data are within the paper.

**Funding:** This study has been supported by the National Institutes of Health grant R35 GM122492

## Abstract

Wilms tumor-1-interacting protein (Wtip) is a LIM-domain-containing adaptor that links cell junctions with actomyosin complexes and modulates actomyosin contractility and ciliogenesis in *Xenopus* embryos. The Wtip C-terminus with three LIM domains associates with the actin-binding protein Shroom3 and modulates Shroom3-induced apical constriction in ectoderm cells. By contrast, the N-terminal domain localizes to apical junctions in the ectoderm and basal bodies in skin multiciliated cells, but its interacting partners remain largely unknown. Targeted proximity biotinylation (TPB) using anti-GFP antibody fused to the biotin ligase BirA identified SSX2IP as a candidate protein that binds GFP-WtipN. SSX2IP, also known as Msd1 or ADIP, is a component of cell junctions, centriolar satellite protein and a targeting factor for ciliary membrane proteins. WtipN physically associated with SSX2IP and the two proteins readily formed mixed aggregates in overexpressing cells. By contrast, we observed only partial colocalization of full length Wtip and SSX2IP, suggesting that Wtip adopts a 'closed' conformation in the cell. Furthermore, the double depletion of Wtip and SSX2IP in early embryos uncovered the functional interaction of the two proteins during neural tube closure. Our results suggest that the association of SSX2IP and Wtip is essential for cell junction remodeling and morphogenetic processes that accompany neurulation. We propose that TPB can be a general approach that is applicable to other GFP-tagged proteins.

## Introduction

The Ajuba family of LIM-domain-containing proteins are adaptors that participate in various processes that involve actomyosin contractility. Ajuba localizes to adherens junctions and modulates Rac1 activity in mammalian cells [1–4]. At the junctions, Ajuba associates with α-catenin and inhibits Hippo signaling in a tension-dependent manner [5–7]. Also, Ajuba homologs localize to the centrosomes of mammalian and Drosophila cells and function in cell division and mitotic spindle orientation [1,8,9].

to SYS. The funders had no role in study design, data collection and analysis, decision to publish, or preparation of the manuscript.

**Competing interests:** The authors have declared that no competing interests exist.

Wilms tumor 1-interacting protein (Wtip) is one of Ajuba homologues. Wtip is composed of two conserved domains that exhibit distinct properties. The C-terminus of Wtip contains three LIM domains and interacts with Shroom3, an actin binding protein that can ectopically induce apical constriction in different cell types [10,11]. Wtip also binds the core planar cell polarity (PCP) protein Prickle3 and has been implicated in PCP and ciliogenesis in *Xenopus* and zebrafish embryos [12–14]. Furthermore, the C-terminal domain of Wtip has been found in the nucleus, where it is proposed to associate with transcriptional repressors of the Snail family and regulate transcription [15]. By contrast, the functions of the Wtip N-terminal domain that includes the first 480 amino acids of the protein (WtipN) are largely unknown. We find that the WtipN localizes to puncta at ectodermal cell junctions and to basal bodies in the epidermal multiciliated cells [14], however, the molecular mechanisms underlying its functions remain to be elucidated.

In this study, we searched for proteins associating with WtipN using a novel proximity biotinylation approach. The promiscuous bacterial biotin ligase BirA* has been fused with the GFP-specific single domain antibody, also known as GFP-binding protein (GBP) [16,17]. This fusion protein (BirA-GBP) would be expected to target the biotinylation activity to the immediate proximity of GFP-WtipN in vivo. This approach identified SSX2IP, also known as Msd1 and ADIP [18], as one of the main candidates. Interestingly, SSX2IP is a maturation factor for mitotic centrosomes [19], and a targeting factor for ciliary membrane proteins cooperating with Cep290, the BBSome, and Rab8 [20]. SSX2IP also interacts with Afadin and α-Actinin [18] and may link cell junctions to the actin cytoskeleton. This interaction of Wtip and SSX2IP has been further validated by the immunoprecipitation and the colocalization of the two proteins in ectoderm cells. We propose that the targeted proximity biotinylation (TPB) approach may help defining protein networks for other GFP-tagged proteins.

## Materials and methods

### Ethics statement

This study was carried out in strict accordance with the recommendations in the Guide for the Care and Use of Laboratory Animals of the National Institutes of Health. The protocol 04–1295 was approved by the IACUC of the Icahn School of Medicine at Mount Sinai.

### Plasmids, mRNA synthesis, morpholinos

WtipN encodes the N-terminal 480 amino acids of Xenopus Wtip. The plasmids for *Xenopus* Wtip including Flag-Wtip, GFP-WtipN, GFP-WtipC, HA-RFP-WtipN, HA-RFP-Wtip, Flag-WtipN, Flag-GFP, GFP-Prickle3C and Flag-Prickle3C have been previously described [12,14]. A plasmid encoding a mutated myc-BirA [21], a gift from Brian Burke, was subcloned into pCS2 vector. WtipN was fused to BirA by subcloning in pCS2-myc-BirA vector to generate WtipN-MycBirA. BirA DNA was fused to the N-terminus of the DNA fragment encoding a single domain antibody specific for GFP [17] in pCS2 vector (BirA-GBP). The insert from the human Msd1/SSX2IP plasmid (a gift from T. Toda) was subcloned into pCS2-GFP vector to make GFP-hSSX2IP. *Xenopus* SSX2IP cDNA clone (SSX2IP.S) was obtained from Dharmacon and the insert was subcloned into pXT7-EGFP to make GFP-XSSX2IP-pXT7. Details of cloning are available upon request. Capped mRNAs were made by in vitro transcription from linearized plasmids as templates with T7 or SP6 RNA polymerases using mMessage mMachine kit (Ambion). Wtip morpholino (WMO) has been described previously (Chu et al., 2016), *Xenopus* SSX2IP MO (SMO) was purchased from GeneTools and had the following sequence: 5'- TAACTCCTCGACTCCTTCTGGACAG-3'.

### *Xenopus* embryo culture, microinjections and neural tube closure analysis

*In vitro* fertilization and culture of *Xenopus laevis* embryos were carried out as previously described [22]. Staging was according to [23]. For microinjections, four-cell embryos were transferred into 3% Ficoll in 0.5 x MMR buffer and 10 nl of mRNAs with or without biotin solution was injected into one or more blastomeres. Amounts of injected mRNA or biotin per embryo have been optimized in preliminary dose-response experiments and are indicated in figure legends.

For neural tube closure analysis, embryos were unilaterally injected with suboptimal doses of Wmo or Smo (10 ng each, as determined in prior studies) or with the mixture of these morpholinos. Neural fold formation in the injected embryos was monitored at neurula stage 18–19 at the time of neural tube closure. The results are presented for three independent experiments.

### Immunoblot analysis

Immunoblotting was carried out as previously described [24]. Briefly, whole embryos were lysed in the lysis buffer containing 1% Triton X-100, 50 mM sodium chloride, 50 mM Tris-HCl at pH 7.6, 1 mM EDTA, 0.6 mM phenylmethylsulphonyl fluoride (PMSF), 10 mM sodium fluoride and 1 mM sodium orthovanadate. Proteins were separated by SDS-polyacrylamide gel electrophoresis and transferred to the PVDF membrane for immunodetection. Antibodies against the following antigens were used: myc (1:200, mouse monoclonal 9E10 hybridoma supernatant), His (1:1000, mouse monoclonal, Invitrogen), Flag M2(1:1000, mouse monoclonal, Sigma), GFP (1:1000, B-2 mouse monoclonal, Santa Cruz). Secondary antibodies were against mouse or goat IgG conjugated to HRP (1:3000, Jackson Immunoresearch or Santa Cruz). Biotinylated proteins were detected similarly with the following modifications. Membranes were blocked in 3% nonfat milk in PBS with 0.1% Tween 20 and incubated in the same buffer with HRP-conjugated streptavidin (1:5,000, Invitrogen).

For co-immunoprecipitation of Flag-tagged Wtip or GFP constructs with GFP-hSSX2IP, 30 embryos at 4–8 cell stages were injected with 300 pg Flag-Wtip, Flag-WtipN or Flag-GFP and 300 pg GFP-hSSX2IP RNA into four animal blastomeres. When control embryos reached stage 11, embryos were lysed in 500 μl of the lysis buffer and cleared by centrifugation at 16,000 g for 4 min. The supernatant was incubated with 2 μl of anti-Flag M2 agarose beads overnight at 4˚C. After washing the beads, 2 x SDS sample buffer was added, heated and the samples were analyzed by immunoblotting.

### Affinity capture of biotinylated proteins

One hundred embryos at the 4-8-cell stage were injected four times into animal blastomeres with 10 nl of RNA (200 ng) mixed with Biotin (1.2 mM). The embryos were cultured until st. 12.5 or st. 14. Then, the embryos were lysed in 1 ml of the lysis buffer and the lysates were centrifuged at 16,000 g for 4 min. Supernatants were incubated with 40 μl neutravidin beads (Invitrogen) at 4˚C overnight. The beads were washed with PBS four times, the bound proteins were eluted by 50 μl of the SDS-sample buffer and heating at 98˚C for 5 min. 10% of the sample was reserved for immunoblotting. For preparative scale, 90% of the sample were loaded on a gel.

### Protein identification by mass spectrometry

Proteins eluted from the avidin beads by boiling in the SDS containing sample buffer and separated by SDS-PAGE using standard procedures [24]. Separated proteins were visualized by

Coomassie Blue staining. The whole gel lanes were cut in several gel bands and submitted to in-gel trypsin digestion and LC-MS/MS analysis at the Keck Proteomics Laboratory (Yale University). Data were annotated using *Xenopus laevis* genome version 9.2 at Xenbase (www. xenbase.org). The results were analyzed by the Scaffold software. The acceptance level for proteins was two identified peptides with minimum 95% probability each.

### Fluorescent protein colocalization in ectoderm explants

Four-to-eight cell embryos were injected with RNAs encoding HA-RFP-WtipN, HA-RFP-Wtip and/or GFP-XSSX2IP (50–300 pg each). When the embryos reached stages 10.5 or 12.5, ectoderm has been dissected, fixed and mounted for imaging. Fluorescent images of protein aggregates were captured using AxioImager microscope (Zeiss) and Axiovision imaging software. Each experimental group had 7–15 embryos for three independent experiments.

## Results

### WtipN is localized to the basal bodies of multiciliated skin cells

We have previously described the subcellular distribution of Wtip in junctional puncta in epidermal ectoderm and neuroectoderm at late gastrula/early neurula stages [14] and at the basal bodies of ciliated endoderm cells of the *Xenopus* gastrocoel roof [12]. We further explored the localization of WtipN, encoding the first 480 amino acids of the protein, in embryos microinjected with GFP-WtipN RNA. In early gastrula ectoderm, GFP-WtipN was detected at cell junctions (**Fig 1A and 1B**). At tailbud stages, GFP-WtipN localized to basal bodies of skin multiciliated cells as well as cell junctions (**Fig 1C**). These findings are consistent with our previous observations [12,14], however the proteins targeting WtipN to the centrosome or cell junctions have not been identified.

### BirA and the BirA-WtipN fusion is active in *Xenopus* embryos

To search for new interacting partners of Wtip, we decided to use proximity biotinylation that allows to isolate associated proteins by labeling them *in vivo* with biotin using a mutated *E. coli* biotin ligase BirA [21]. We generated a fusion of BirA with WtipN (myc-BirA-WtipN) and tested its enzymatic activity in *Xenopus* embryos. Early embryos were injected with myc-BirA or myc-BirA-WtipN RNA with or without biotin and cultured until neurula stages. We observed robust self-biotinylation of both myc-BirA-WtipN and myc-BirA in the presence of biotin, confirming that BirA is active both on its own and as a fusion (**Fig 1D**). Notably, in the absence of biotin, the background was low, except for two major protein bands around 70 and 120 kDa recognized by Streptavidin-HRP, possibly representing endogenous biotin-containing proteins. Pulldowns with neutravidin beads contained many additional bands in the sample from myc-BirA-WtipN-expressing embryos (**Fig 1E**), presumably corresponding to WtipN-interacting proteins. These results suggest that BirA can be used directly for proximity biotinylation in *Xenopus*, complementing previous observations for this and other models [25–28].

### Targeted proximity biotinylation

We next modified the approach by fusing BirA to the coding sequence of a single domain antibody against GFP, or GFP-binding protein [16,17]. Such fusion (BirA-GBP) would be predicted to biotinylate any GFP-tagged protein and other proteins in its immediate proximity. This construct would serve as a universal tool for targeted proximity biotinylation (TPB). To

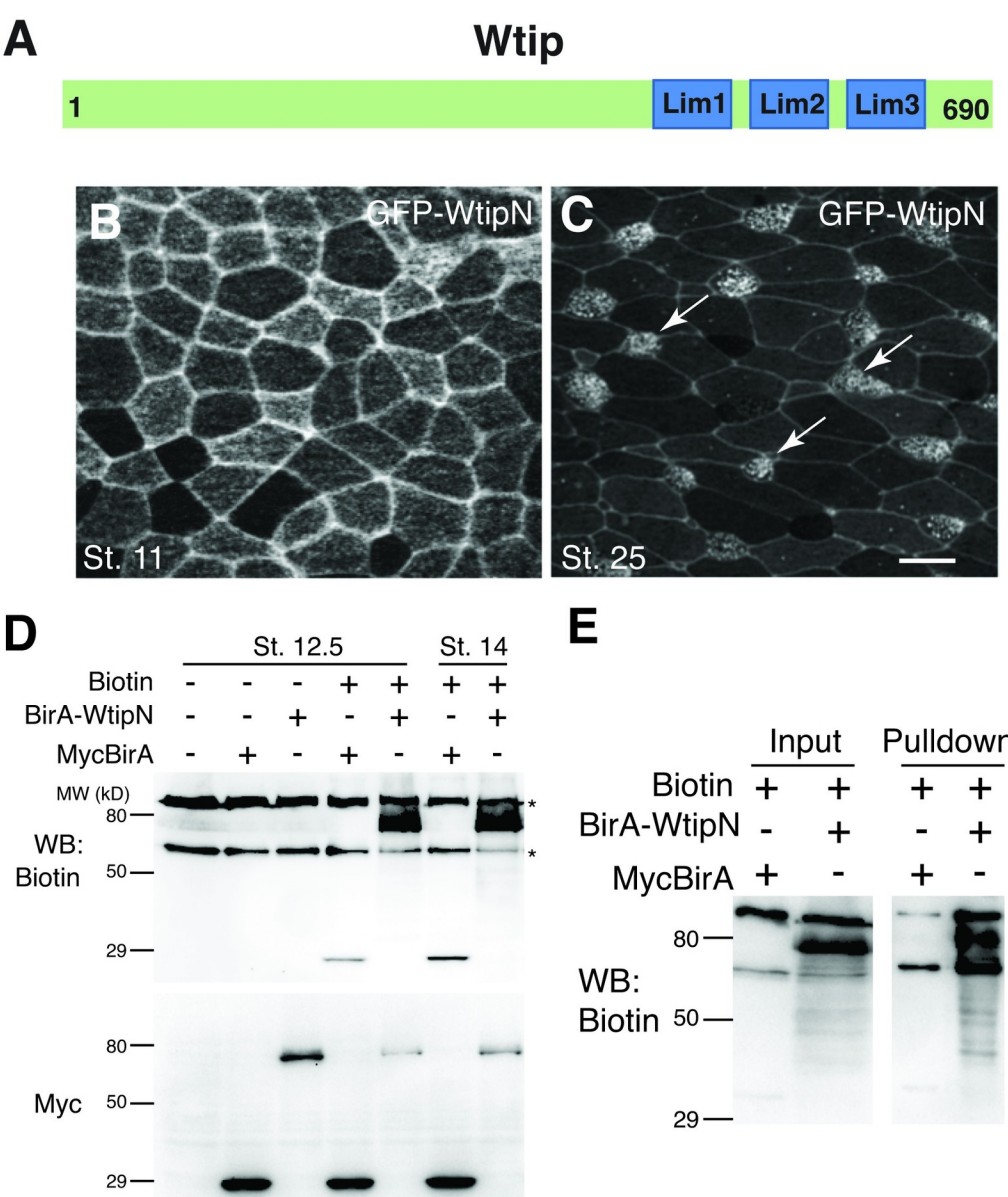

**Fig 1. Biotinylation of the BirA-WtipN fusion in Xenopus embryos.** A, Schematic of Wtip. B, C, Embryos were injected with 200 pg of GFP-WtipN RNA, fixed and imaged at the indicated stages. B, GFP-WtipN junctional staining in stage 11 ectoderm. C, GFP-WtipN is localized to basal bodies of skin multiciliated cells (arrows, stage 25). Scale bar, 20 μm. D, Biotinylation of the BirA-WtipN fusion. Four-cell embryos were injected with myc-BirA RNA (300 pg) or mycBirA-WtipN RNA (700 pg) with or without 0.8 mM biotin. Self-biotinylation of mycBirA and mycBirA-WtipN is revealed only in the presence of biotin. Protein levels were assessed with anti-myc antibody. Asterisks indicate endogenous avidin-binding proteins. E. Biotinylated proteins were pulled down with neutravidin beads. In addition to self-biotinylated mycBirA-Wtip, multiple biotinylated bands are detected in the lysates as well as in pulldown.

validate the efficiency and specificity of this approach, we injected BirA-GBP RNA into early embryos together with GFP-WtipN and Flag-WtipN.

As expected, we observed an efficient and specific biotinylation of GFP-WtipN but not Flag-WtipN, when these were coexpressed with BirA-GBP and biotin (**Fig 2A**). Of note, Bir-A-GBP failed to biotinylate Flag-GFP, possibly due to lack of suitable lysine residues (**Fig 2A**). To show that TPB can be applied to a different protein, we confirmed that BirA-GBP

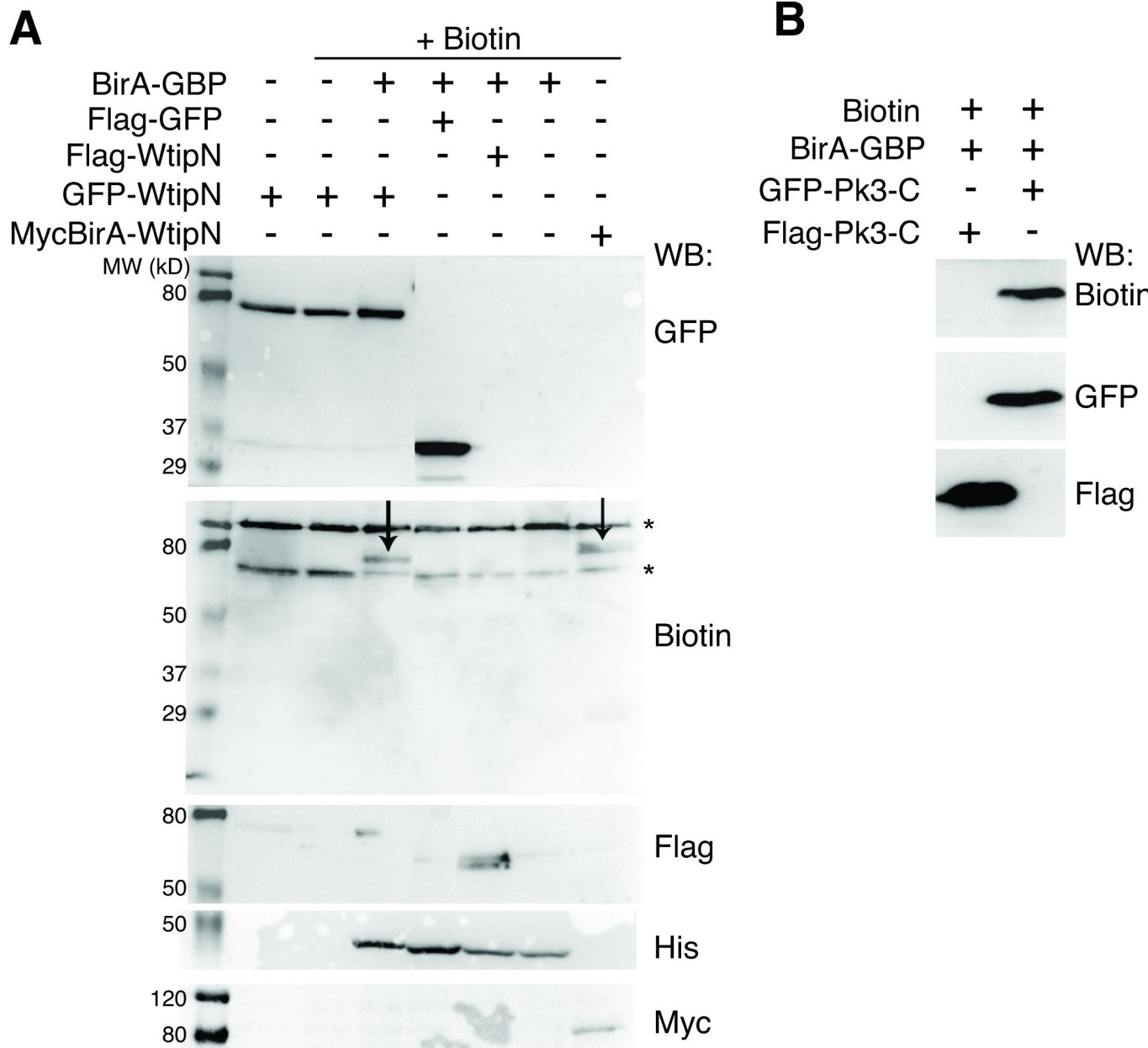

**Fig 2. Specificity of BirA-GBP-mediated biotinylation.** A, GFP-WtipN but not Flag-WtipN is biotinylated (left arrow) by BirA-GBP in the presence of biotin. MycBirA-WtipN is a positive control for biotinylation (right arrow). B, GFP-Prickle3C protein, but not Flag-Prickle3C, was biotinylated by BirA-GBP. Protein expression levels are assessed with tag-specific antibodies as indicated (A, B).

biotinylates GFP-Prickle3C but not Flag-Prickle3C [12] (**Fig 2B**). These results suggest that TPB may be useful for protein network analysis for GFP-tagged proteins in general.

## Identification of SSX2IP/Msd1/ADIP as a novel interacting partner of WtipN

To search for proteins interacting with WtipN, we carried out TPB *in vivo*, in embryos coexpressing GFP-WtipN and BirA-GBP in the presence of biotin (**Fig 3A**). Embryos coexpressing

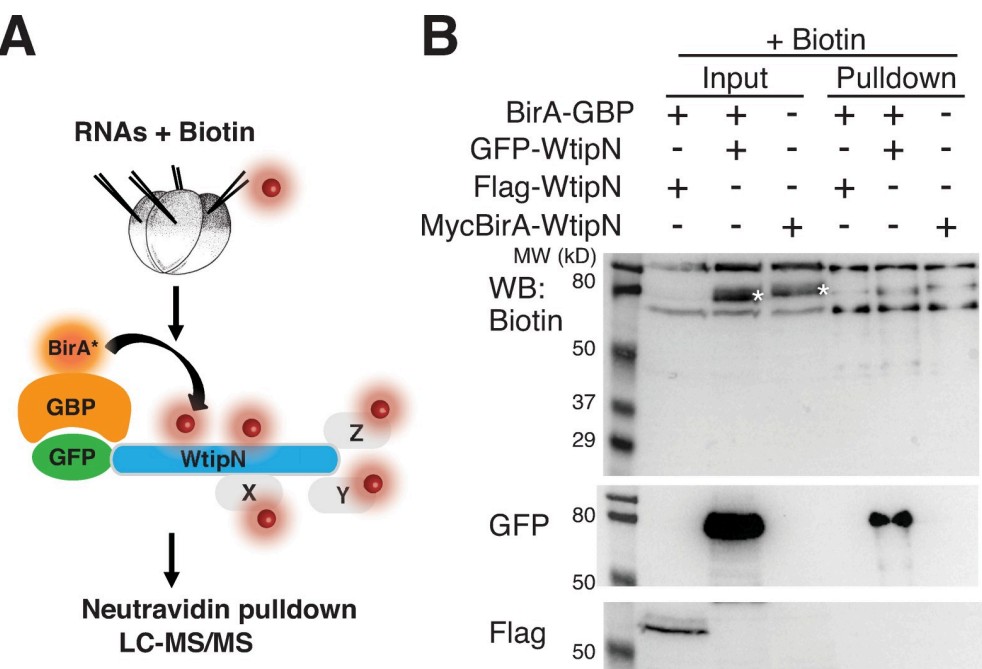

C

| Identified proteins | Scaffold score | | |
|---|---|---|---|
| | Myc-BirA-WtipN | Flag-WtipN + BirA-GBP | GFP-WtipN + BirA-GBP |
| Ssx2ip.L | 13 | - | 8 |
| Ssx2ip.S | 12 | - | 7 |
| Lmnb1 | 3 | - | 7 |
| Hspa9 | 3 | - | 5 |
| Rnf112.1.S | 3 | - | 5 |
| Me2 | 2 | - | 5 |
| Faf2.L | 1 | - | 4 |
| Nupl1 | 2 | - | 4 |
| Dnm1l | 1 | - | 3 |
| Epcam.L | 4 | - | 3 |
| Cdkal1 | 4 | - | 2 |
| Pof1b | 6 | - | 2 |

**Fig 3. Identification of SSX2IP by targeted proximity biotinylation.** A, Scheme of targeted proximity biotinylation (TPB) with GFP-WtipN. Early embryos were injected with indicated RNAs (200 pg each) and biotin (1.2 mM). B, Biotinylated proteins were enriched with neutravidin agarose beads and immunoblotted with indicated antibodies. Biotinylated GFP-WtipN (asterisk) but not Flag-WtipN in the presence of BirA-GBP was detected in the pulldown. MycBirA-WtipN (asterisk) was biotinylated in the absence of BirA-GBP. C, Top hits identified by SCAFFOLD after LC-MS/MS analysis in both experimental groups but not in the negative control group.

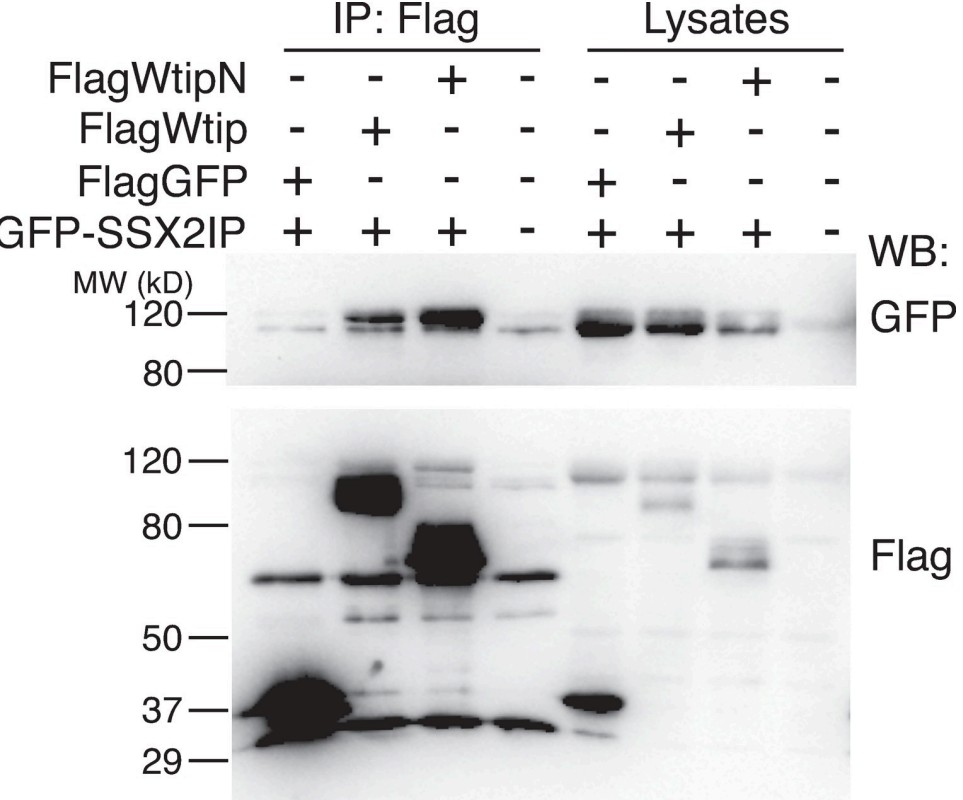

**Fig 4. The physical association of SSX2IP and Wtip.** Four-cell embryos were coinjected with GFP-hSSX2IP RNA and either Flag-WtipN or Flag-Wtip RNA (300 pg each). Flag-GFP RNA (300 pg) was the negative control. Embryos were lysed at stage 11 and lysates were immunoprecipitated with anti-Flag agarose beads and immunoblotted with antibodies specific for Flag and GFP. WtipN coprecipitated with SSX2IP more efficiently than full length Wtip.

Flag-WtipN and BirA-GBP were used as negative controls (**Fig 3B**). A separate group of embryos was injected with Myc-BirA-WtipN. Biotinylated proteins were captured with neu-travidin-agarose beads, the bound proteins were separated by SDS-PAGE and visualized by Simple Blue staining. Gel slices were analyzed by LC-MS/MS to identify candidate interacting proteins. This analysis identified several proteins that were present in the group of GFP-WtipN and BirA-GBP, but absent in the group with Flag-WtipN and BirA-GBP (**Fig 3C**). Multiple identified peptides belonged to the *Xenopus* homologue of SSX2IP/ADIP/Msd1 [18,19,29]. The same protein was identified at high confidence using Myc-BirA-WtipN as a bait (**Fig 3C**). These findings indicate that SSX2IP and WtipN interact.

## SSX2IP associates with WtipN in embryonic ectoderm

We next validated the identified interaction by co-immunoprecipitation (**Fig 4**). Flag-WtipN protein strongly associated with GFP-SSX2IP protein at the beginning of gastrulation when co-expressed in *Xenopus* embryos. Of note, the full length Wtip protein was less efficient in the binding of GFP-SSX2IP protein (**Fig 4**). These experiments strongly suggest that SSX2IP associates with WtipN *in vivo*. Consistent with this possibility, both Wtip and SSX2IP have been found at or near the basal bodies [12,14,20,30].

We next evaluated the subcellular distribution of SSX2IP in ectodermal cells. In embryos injected with low doses of GFP-SSX2IP RNA (50 pg), SSX2IP is distributed to puncta adjacent to the cell cortex at the beginning of gastrulation (**Fig 5A and 5B**). By the end of gastrulation,

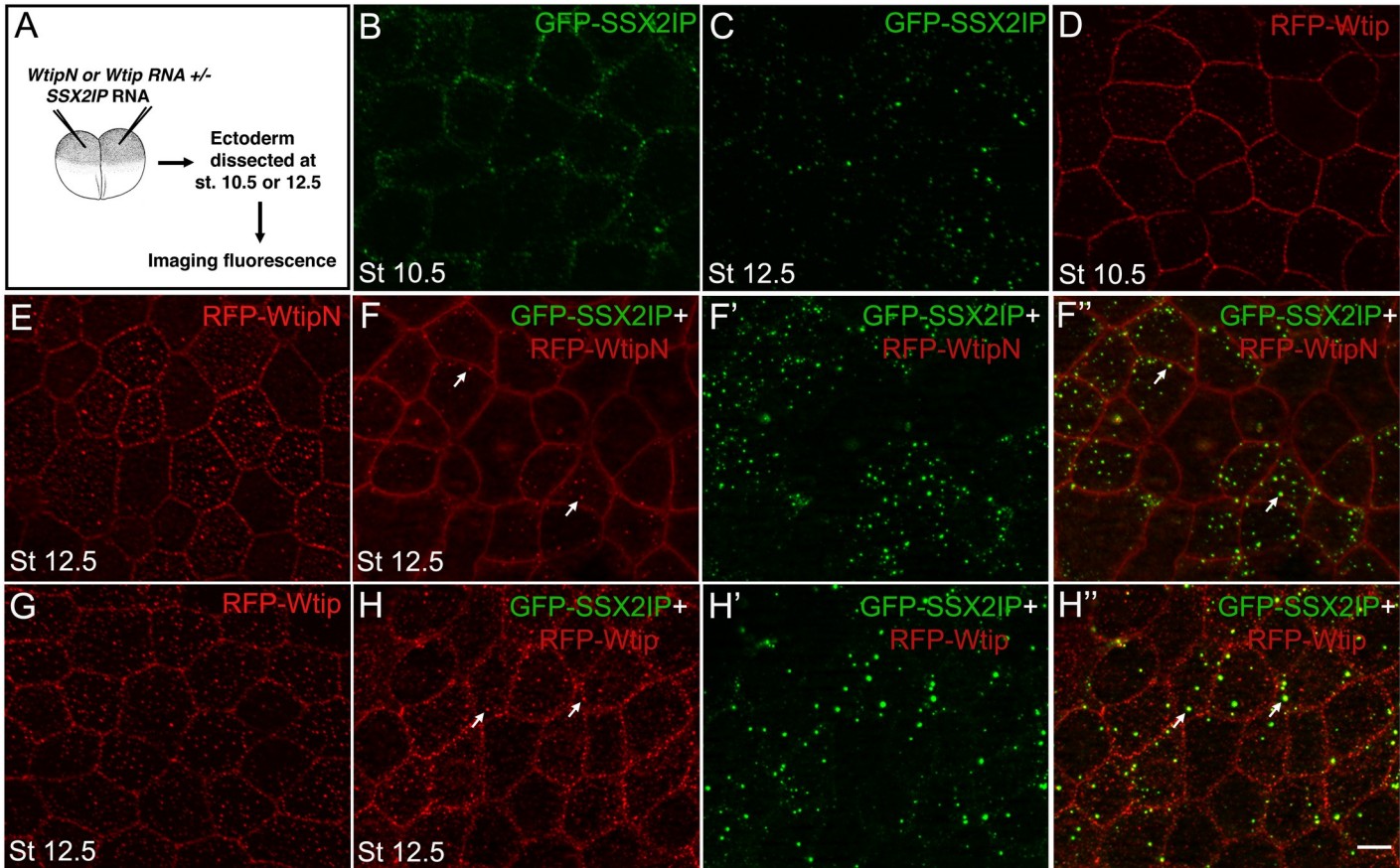

**Fig 5. SSX2IP and WtipN co-aggregate in ectoderm cells.** A, Experimental scheme. HA-RFP-Wtip or HA-RFP-WtipN RNA (50 pg each) was coinjected with GFP-SSX2IP RNA (50 pg) into four-cell embryos. When the embryos reached stage 10 or 12.5, ectoderm was dissected, fixed, and fluorescent protein localization at the cortex and cytoplasmic puncta was imaged at indicated stages. B, GFP-SSX2IP, stage 10.5. C, GFP-SSX2IP, stage 12.5. D, HA-RFP-Wtip, stage 10.5. E, HA-RFP-WtipN, stage 12.5. F-F", Mixed cytoplasmic aggregates (arrows) in cells coexpressing GFP-SSX2IP and RFP-WtipN, stage 12.5. F, red channel, F' green channel, F", merged image. G, HA-RFP-Wtip, stage 12.5. H-H", Partial colocalization of GFP-SSX2IP and HA-RFP-Wtip in cytoplasmic puncta (arrows). Scale bar, 15 μm. Images are representative of 3 independent experiments, with 10–15 embryos imaged per group in each experiment.

SSX2IP accumulated in spherical cytoplasmic aggregates of variable size (**Fig 5C**). Given that Wtip and WtipN also form cytoplasmic and cortical aggregates (**Fig 5D, 5E and 5G**), we asked whether they would colocalize with SSX2IP. We observed that GFP-SSX2IP and HA-RFP-WtipN puncta readily colocalized in the cytoplasm of ectoderm cells (**Figs 5F–5F"** and **S1**). The SSX2IP/WtipN protein complexes increased in size at higher doses of RNAs (**S1 Fig**). Notably, full length HA-RFP-Wtip showed only limited colocalization with SSX2IP (**Fig 5H–5H"**). Together these experiments suggest that SSX2IP binds the N-terminal fragment of Wtip and likely modulates its function at the same subcellular location, such as the centrosome or cell junctions.

## Functional interaction of SSX2IP and Wtip during neural tube closure

Since both Wtip and SSX2IP associate with cell junctions and Wtip has been implicated in neural tube closure [14,18], we wanted to test whether the two proteins functionally interact. We therefore evaluated the phenotype of the morphants depleted of both Wtip and SSX2IP using translation-blocking MOs. Injection of moderate doses (10–30 ng) of SSX2IP-specific MO caused neural tube closure defects consistent with the suspected role of SSX2IP in

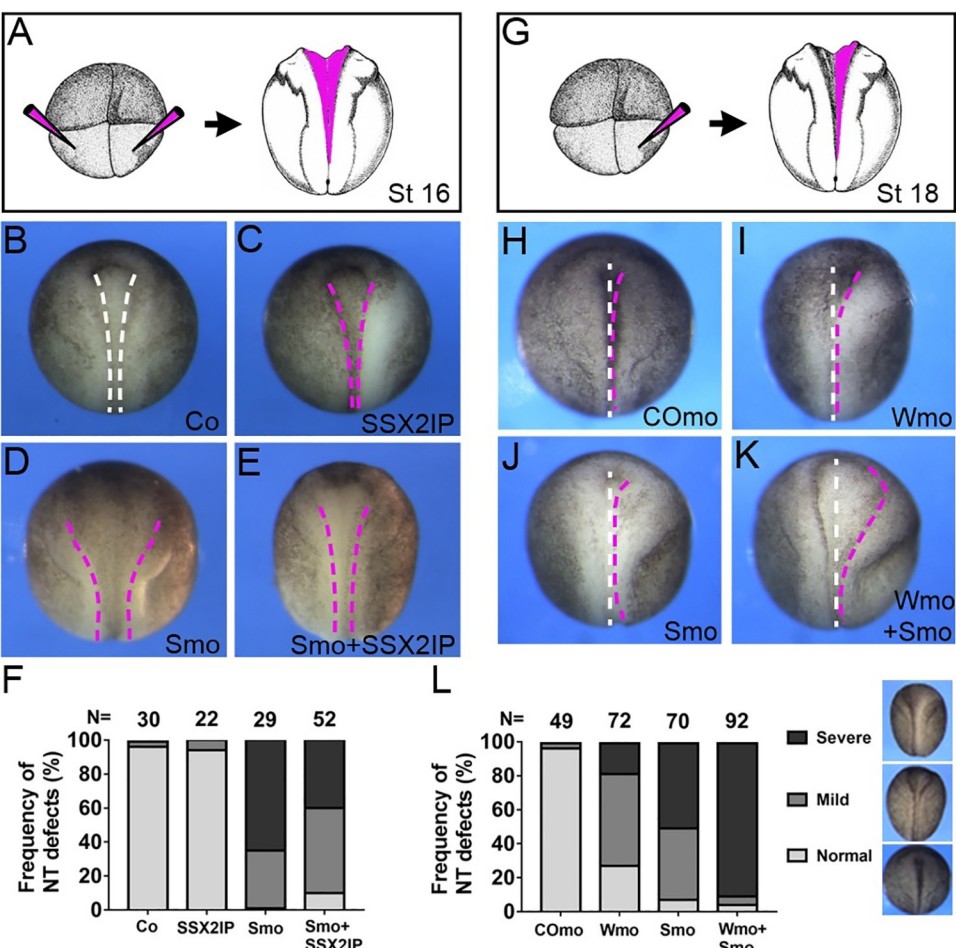

**Fig 6. SSX2IP and Wtip functionally interact to promote neural tube closure.** Embryos were injected bilaterally or unilaterally with 10 ng of Wtip MO (Wmo) or SSX2IP MO (Smo) or 20 pg of human SSX2IP RNA as indicated. Neural groove formation along the anteroposterior axis was assessed at stages 16–18. A-F, Neural tube closure defects in SSX2IP morphants. A, Scheme of the experiment. B-D, Dorsal views of representative embryos at stages 16–17, anterior is up. B, Control uninjected embryo, C, SSX2IP RNA injection. D, Embryo injected with Smo. E, Embryo injected with Smo and SSX2IP RNA. Human SSX2IP RNA partly rescues neural closure defects. Dashed lines indicate neural fold borders. F, Frequency of neural tube (NT) defects. Data are representative of two experiments. G-L, Functional interaction of SSX2IP and Wtip. G, Scheme of the experiment. H-L, Representative embryos at stages 18–19 injected with control MO (CoMO, H), Wmo (I), Smo (J) or both Wmo and Smo (K). Midline, white dashed line, neural fold border, magenta dashed line. Anterior is up. L, Frequency of neural tube (NT) defects in the morphants. Data represent three different experiments. In F and L, total number of embryos per group is shown above each bar. Mild, severe and normal phenotypes have been scored separately, with examples shown in L.

neurulation. This phenotype has been partly rescued by the coinjection of human SSX2IP RNA, although we noticed considerable variability in this assay in different experiments (**Fig 6A–6F**). Whereas each MO had only a mild effect on neural tube closure, neural folding has been severely affected in embryos injected with both MOs (**Fig 6G–6L**). Our observations indicate that SSX2IP and Wtip functionally interact in neural tube closure.

## Discussion

In this study, we used targeted proximity biotinylation to search for new interaction partners of Wtip, a LIM-domain protein of the Ajuba/Zyxin family. Wtip is present in many tissues and has been shown to localize to the centrosome/basal bodies and cell junctions [12,14]. This

diverse localization may underlie the proposed functions of Wtip in morphogenesis and planar cell polarity as well as its roles in the centrosomal organization and ciliogenesis [12–14]. Wtip has been also implicated in forming a complex with the Snail transcription factors leading to the activation of neural crest markers [15]. The knowledge of new interaction partners of Wtip should help elucidate how it functions in the cell.

Proximity biotinylation or BioID (biotin identification) is a useful method for detecting weak and transient protein interactions under physiological conditions. This approach over-comes non-specific contamination that often accompanies traditional pulldowns and is more sensitive and specific as compared to conventional immunoprecipitations or yeast-two-hybrid screens [31]. We modified this technique by fusing BirA with GBP, an anti-GFP antibody that brings the enzyme into close proximity with GFP-WtipN. One advantage of our approach is that the generation of the direct fusion of the 'bait' protein with BirA is no longer needed. Also, this method may identify a more diverse protein population due to the increased distance or spatial flexibility between the enzyme and the bait. Importantly, SSX2IP has been detected by targeting BirA-GBP to GFP-WtipN and by the direct fusion protein WtipN-BirA, increas-ing our confidence in the result. Based on our results and a similar study using zebrafish embryos [32], we propose that TPB is a versatile general technique that is applicable for protein interaction studies of GFP-tagged proteins. However, lack of biotinylation of Flag-GFP in our experiments suggests possible sterical hindrance of the epitope or lack of lysine residues needed for biotinylation, indicating that not all proteins are suitable targets for this approach.

Our analysis identified SSX2IP/ADIP/Msd1 as a protein associating with the N-terminus of Wtip. SSX2IP is a conserved centriolar satellite protein that anchors microtubules to the cen-trosome [29] and functions in ciliogenesis [30]. We validated the binding of WtipN to SSX2IP using immunoprecipitation and *in vivo* colocalization of the two proteins. The implication of both Wtip and SSX2IP in ciliogenesis [13,30] suggests a potential role of this interaction in PCP signaling. Notably, upon coexpression, WtipN and SSX2IP formed mixed spherical cyto-plasmic aggregates of the size dependent on the dose of injected RNAs. By contrast, full length Wtip only partly colocalized with SSX2IP, suggesting that the protein might be present in a closed conformation caused by the interaction of the N- and C-terminal fragments [14]. The aggregation of WtipN and SSX2IP is similar to known cases of liquid phase separation that has been observed for the centrosome assembly both *in vitro* and *in vivo* [33–35]. The physiologi-cal significance of these observations and the mechanism regulating Wtip conformational changes remain to be clarified in future studies.

Like SSX2IP, Wtip is present at the centrosome and basal bodies of different cell types and appears to require its N-terminus for this localization [12,14]. Thus, the interaction of Wtip with SSX2IP may be a prerequisite for the centrosomal/basal body localization for one or the other protein or may be relevant for the assembly of the pericentriolar material. Besides the roles at the centrosome and/or basal body, the association of Wtip and SSX2IP may play a role in actomyosin dynamics during junction remodeling. Both proteins interact with tension-sen-sitive molecules that organize actomyosin complexes at cell junctions [14,18]. Consistent with the roles at cell junctions, we observed the functional interaction of the two proteins during neural tube closure. Further analysis is needed to demonstrate how SSX2IP contributes to reg-ulation of actomyosin contractility.

## Supporting information

**S1 Fig. Colocalization of SSX2IP and WtipN in ectoderm cells.** A, Experimental scheme. RFP-WtipN RNA (300 pg) and GFP-XSSX2IP RNA (300 pg) were coinjected into four-cell embryos. When the embryos reached stage 12.5, they were fixed, and the ectodermal tissue

was imaged. B, GFP-SSX2IP localization in ectoderm cells. SSX2IP colocalizes with WtipN in cytoplasmic puncta. C, RFPWtipN localization at the junctions and cytoplasmic puncta. D-D", Coexpression of both proteins revealed mixed cytoplasmic aggregates (arrow). Scale bar, 15 μm. Data are representative of 3 independent experiments.
(PDF)

**S2 Fig. Full gel images.**
(PDF)

# Acknowledgments

We thank T. Toda for the hMsd1 plasmid. We also thank Andriani Ioannou and Fei Wu for their contributions at the early stages of this work, Miho Matsuda for advice and help with imaging of fluorescent protein localization, and members of the Sokol laboratory for discussions. We acknowledge the help of the Keck Proteomics Laboratory in protein identification by LS-MS/MS.

# Author Contributions

**Conceptualization:** Alice H. Reis, Bo Xiang, Sergei Y. Sokol.

**Data curation:** Olga Ossipova.

**Formal analysis:** Bo Xiang, Keiji Itoh, Sergei Y. Sokol.

**Funding acquisition:** Sergei Y. Sokol.

**Investigation:** Alice H. Reis, Bo Xiang, Keiji Itoh.

**Methodology:** Alice H. Reis, Bo Xiang, Olga Ossipova.

**Software:** Olga Ossipova.

**Supervision:** Sergei Y. Sokol.

**Validation:** Alice H. Reis, Bo Xiang, Keiji Itoh.

**Visualization:** Olga Ossipova.

**Writing – original draft:** Alice H. Reis, Bo Xiang, Sergei Y. Sokol.

**Writing – review & editing:** Olga Ossipova, Keiji Itoh.

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
