## [Decision Letter · Decision Letter 0]

11 Aug 2021

PONE-D-21-20125

Identification of SSX2IP as a Wtip-binding partner bytargeted proximity biotinylation

PLOS ONE

Dear Dr. Sokol,

Thank you for submitting your manuscript to PLOS ONE. After careful consideration, we feel that it has merit but does not fully meet PLOS ONE’s publication criteria as it currently stands. Therefore, we invite you to submit a revised version of the manuscript that addresses the points raised during the review process.

Please perform co-localization with full length Wtip

Please try to use using lower amounts of mRNA injected (say 4x50, max 4x100pg) to reveal a more physiological distribution/co-localization.

Please try to rescue MO phenotypes by co-injection of the corresponding mRNA.

Please revise the use of synergy to describe the effect of Wtip - SSX2IP co-depletion

We look forward to receiving your revised manuscript.

Kind regards,

Claude Prigent

Academic Editor

PLOS ONE

Reviewers' comments:

Reviewer's Responses to Questions

**Comments to the Author**

1. Is the manuscript technically sound, and do the data support the conclusions?

Reviewer #1: Yes

2. Has the statistical analysis been performed appropriately and rigorously? 

Reviewer #1: N/A

3. Have the authors made all data underlying the findings in their manuscript fully available?

Reviewer #1: Yes

4. Is the manuscript presented in an intelligible fashion and written in standard English?

Reviewer #1: Yes

5. Review Comments to the Author

Reviewer #1: see attachment, I paste here only the first paragraph

This is an interesting report of a new interaction between the limb domain adaptor protein Wtip (Ajuba family) implicated at the interface between the actin cytoskeleton and cell junctions, and also implicated in ciliogenesis, and SSX2IP, a protein that involved in microtubule-centrosome interactions. Specifically, the authors were looking for interactors of the N-term of Wtip, which they had previously found to be crucial for localization both at cell-cell junctions and at cilia basal bodies..................

6. PLOS authors have the option to publish the peer review history of their article (what does this mean?). If published, this will include your full peer review and any attached files.

Reviewer #1: No

---

## [Author Response · Author response to Decision Letter 0]

13 Sep 2021

PONE-D-21-20125. Response to Reviews.

Identification of SSX2IP as a Wtip-binding partner by targeted proximity biotinylation

We first wish to thank the referee and the editor for the constructive comments that helped improve our manuscript.

Point-by-point responses to referee 1

This is an interesting report of a new interaction between the limb domain adaptor protein Wtip (Ajuba family) implicated at the interface between the actin cytoskeleton and cell junctions, and also implicated in ciliogenesis, and SSX2IP, a protein that involved in microtubule-centrosome interactions. Specifically, the authors were looking for interactors of the N-term of Wtip, which they had previously found to be crucial for localization both at cell-cell junctions and at cilia basal bodies.

The interaction was first identified by biotinylation, pulldown and proteomics, using a protocol based on fusing an anti-GFP nanobody to the biotin ligase BirA. The authors comment on the utility of this new protocol, as it avoids fusing each bait protein with BirA, as done in standard biotinylation methods. One should note, however, that this method has its own caveats, as it failed to biotinylate a simple Flag-tagged GFP. This in my view hint at steric hindrance and/or epitope marking, which may depend on the type of GFP-fusion protein. One would wish to see this caveat included in the discussion.

We discuss this issue in the amended text (pp. 14, top) as suggested by the reviewer.

As for the interaction Wtip - SSX2IP, the interaction appears robust, as validated by co-immunoprecipitation. How direct/indirect is the interaction was not investigated. A simple protein staining (silver stain?) of an SDSPAGE of the material retrieved from the biotin pulldown may be a helpful indicator of how many additional, non-biotinylated potential partners were fished.

The Coomassie Blue SDS-PAGE gel used for the pull-downs contained many background bands in both control and experimental samples but did not reveal specific protein bands of low abundance. We therefore decided not to include this gel into the manuscript.

The authors also validate the interaction by co-expression and immunofluorescence, which reveals a very convincing, extensive co-localization in cytoplasmic blobs. As expected from literature, SSX2IP is absent from the cell junctions, however one may have hoped that the Wtip-SSX2IP colocalization would happen in basal bodies. 

I have here two comments: 

1) Co-localization should also be performed using full length Wtip, not only its N-terminus.

Prompted by the reviewer, we expressed full-length Wtip to determine whether it co-distributes with SSX2IP. Whereas WtipN and SSX2IP colocalized in many cytoplasmic puncta, full-length Wtip exhibited only partial colocalization in this assay. This finding is consistent with our immunoprecipitation data (Fig. 4) and supports a model, in which the excess of Wtip adopts a ‘closed’ conformation in the cell. These results are now presented in the new Fig. 5, and discussed on pp. 11-12, and p. 14, bottom. 

2) The large number and the size of the blobs can be obviously explained by the high level of overexpression. The amount of mRNA injected (4x300pg for each of the component) is very high. It is fully justified for the pulldown as well as validation of the interaction, but using lower amounts (say 4x50, max 4x100pg) would be likely to reveal a more physiological distribution/co-localization.

As suggested by the reviewer, we injected lower doses (50 pg) of RNAs to study GFP-SSX2IP, RFP-Wtip and HA-RFP-WtipN protein localization during gastrulation. At lower doses of RNAs, the puncta of Wtip and SSX2IP were smaller in size (compare new Fig. 5 and Suppl. Fig. 1). In early gastrulae (stage 10.5), SSX2IP and Wtip were found in puncta near the cortex, but the puncta became more cytoplasmic at stage 12.5 (new Fig. 5).

Although the observed puncta may be non-physiological, this assay allows us to demonstrate the interaction of the two ectopic proteins in the tissue. In this case, the ectopic location is critical for establishing the interaction. Conversely, the co-distribution to basal bodies, in which Wtip and SSX2IP are normally found, would not prove the physical binding.

Finally, two comments on the functional tests using morpholinos:

1) The standard control for specificity of MO phenotypes is a rescue by co-injection of the corresponding mRNA. This is missing here. Of course, one may not expect full rescue, as dosage + spatial targeting could be tricky, yet even partial rescue would be OK.

We have done the experiment and have shown that human SSX2IP/Msd1 partly rescued neural tube closure defects of SSX2IP morphants. This is shown in revised Fig. 6 and discussed on p. 12, bottom.

2) Wtip - SSX2IP co-depletion: It is here dangerous to talk about “synergy”. The results of these co-depletions could simply be explained by an additive effect. If the authors wish to argue for a functional interaction, they may want to decrease the doses of each morpholino, down to levels that show little to no phenotype when injected alone. A strong double-knock down phenotype would then be a better argument for synergistic effect.

The text has been amended to remove the description of the functional Wtip and SSX2IP interactions as ‘synergistic’. 

Responses to the editorial comments

In the revised version of the manuscript, we have addressed the editor’s points that largely reiterate the concerns of reviewer 1.

--Please perform co-localization with full length Wtip.

As requested, we expressed full-length Wtip to determine whether it co-distributes with SSX2IP. Whereas WtipN and SSX2IP colocalized in many cytoplasmic puncta, full-length Wtip exhibited only partial colocalization in this assay (new Fig. 5, text on pp. 11-12, 14, bottom).

--Please try to use lower amounts of mRNA injected (say 4x50, max 4x100pg) to reveal a more physiological distribution/co-localization.

At lower, more physiological doses of RNAs (50 pg each), SSX2IP and Wtip were visible as cortical or cytoplasmic puncta at the beginning of gastrulation (stage 10.5), whereas the SSX2IP puncta were predominantly cytoplasmic at a later stage (stage 12.5) (new Fig. 5, Fig. S1). We acknowledge that the observed aggregates may be non-physiological, however, this assay allows us to demonstrate the interaction of the two ectopic proteins in the tissue. 

--Please try to rescue MO phenotypes by co-injection of the corresponding mRNA.

We found that human SSX2IP partly rescued neural tube closure defects of SSX2IP morphants. This is shown in revised Fig. 6.

--Please revise the use of synergy to describe the effect of Wtip - SSX2IP co-depletion.

The description of the functional interactions as ‘synergistic’ has been removed.

---

## [Decision Letter · Decision Letter 1]

12 Oct 2021

Identification of the centrosomal maturation factor SSX2IP as a Wtip-binding partner by targeted proximity biotinylation

PONE-D-21-20125R1

Dear Dr. Sokol,

We’re pleased to inform you that your manuscript has been judged scientifically suitable for publication and will be formally accepted for publication once it meets all outstanding technical requirements.

Kind regards,

Claude Prigent

Academic Editor

PLOS ONE

Additional Editor Comments (optional):

Reviewers' comments:

Reviewer's Responses to Questions

**Comments to the Author**

1. If the authors have adequately addressed your comments raised in a previous round of review and you feel that this manuscript is now acceptable for publication, you may indicate that here to bypass the “Comments to the Author” section, enter your conflict of interest statement in the “Confidential to Editor” section, and submit your "Accept" recommendation.

Reviewer #1: All comments have been addressed

2. Is the manuscript technically sound, and do the data support the conclusions?

Reviewer #1: Yes

3. Has the statistical analysis been performed appropriately and rigorously? 

Reviewer #1: Yes

4. Have the authors made all data underlying the findings in their manuscript fully available?

Reviewer #1: Yes

5. Is the manuscript presented in an intelligible fashion and written in standard English?

Reviewer #1: Yes

6. Review Comments to the Author

Reviewer #1: The authors have satisfactorily answered to the reviewers comments, by including additional data and/or modifying the text

No further comment

7. PLOS authors have the option to publish the peer review history of their article (what does this mean?). If published, this will include your full peer review and any attached files.

Reviewer #1: No

---

## [Editor Report · Acceptance letter]

20 Oct 2021

PONE-D-21-20125R1 

Identification of the centrosomal maturation factor SSX2IP as a Wtip-binding partner by targeted proximity biotinylation 

Dear Dr. Sokol:

I'm pleased to inform you that your manuscript has been deemed suitable for publication in PLOS ONE. Congratulations! Your manuscript is now with our production department. 

Kind regards, 

on behalf of

Dr. Claude Prigent 

Academic Editor

PLOS ONE